# A Light Spectrometer Device for Crop Disease Monitoring

Joshua J Dhikusooka[1], Ephraim Nuwamanya[2], Estefania Talavera[3], and Godliver Owomugisha [4]

[1]Department of Computer Science, Makerere University, Uganda
[2]National Crops Resources Research Institute, P.O Box 7084, Kampala, Uganda
[3]Faculty of Electrical Engineering, University of Twente, P.O. Box 217 7500 AE Enschede, NL
[4]Faculty of Engineering, Busitema University, P.O. Box 236, Tororo, Uganda

## Abstract

Portable devices for the early detection of crop diseases are needed to support the farmers working in the field. Spectrometers showed their potential in the detection of crop diseases. However, high interpretation skills are needed to use the currently available spectrometers. In this project, we propose a portable device that obtains a spectrum wavelength of 700 nanometers describing the information of the crop. The output of this tool is integrated into a smartphone in the form of an app, making it accessible for use in the field in real applications.

## 1 Introduction

Different types of spectrometry and spectrometers are available in the market (Monakhova et al., 2020). However, most of them are expensive and also require high interpretation skills to use them. In our project, we aim at addressing these limitations by building a portable spectroscopy device. This device is composed of a smartphone add-on spectrometer integrated with a software application for analyzing crop information for field diagnosis. Previous findings by (Owomugisha et al., 2021) found that the most relevant spectral band on a wavelength corresponding to disease detection in cassava plants is between 500 - 600 nms. Therefore, the current work is based on this evidence to build a robust low-cost spectrometer that farmers can use in their field to perform diagnosis at an early stage(Stuart et al., 2021), (Omara et al., 2023). Most spectrometers have been categorized to range from 400-700 nanometer wavelengths and commercially produced spectrometers slightly range up to 1000 nms (Shailesh et al., 2016). The low-cost device developed in the work achieved a wavelengths of 700nms representing information captured from diseased and healthy crops. The portable device has been designed at a cost of ($5 - 8), in contrast to the expensive off-the-shelf spectrometer (approximately $1000).

Ordinarily, disease detection in plants happens through visual symptoms when the plant has critically been damaged. Our hypothesis is that different diseases affect the metabolism of plants differently thus changes in light absorption properties. The tool design follows initial prototype concepts [1] and (Owomugisha et al., 2020a) implemented through a diffraction grating, where light passing through a leaf is split by using a DVD as a low-cost alternative to optical fiber and more powerful light sources. In this work, use a smartphone camera as a receiver for light that has been absorbed from the leaf sample. The improvements in the current design are twofold. Firstly, the development of software that analyzes the light and transforms the acquired image spectrum into a spectrogram, as shown in Figure 1. Secondly, the design of a portable device powered by a smartphone, making it durable for field use over longer periods. The system also includes a mobile application, making it user-friendly for smallholder farmers to collect data or identify crop diseases in the field.

---

*Correspondence: Godliver Owomugisha, ogodliver@eng.busitema.ac.ug

[1]https://publiclab.org/tag/spectrometry

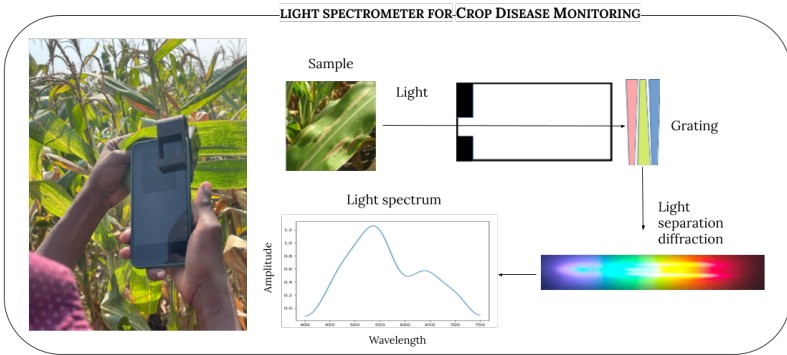

Figure 1: The developed smartphone-assisted spectrometer for crop disease identification.

## 2 EXPERIMENTAL FRAMEWORK

**Data Collection**. Using the developed device, we obtained two types of data: (i) a raw spectral dataset consisting of 360 data points that encompass three distinct classes: Healthy (HLT), Cassava Mosaic Disease (CMD) and Cassava Brown Streak Disease (CBSD). (ii) corresponding leaf images captured using a smartphone camera with a resolution of 720 x 1600 pixels. The plants were visually evaluated and scored under the supervision of an agricultural expert. This dataset is available here [2]

**Results and findings.** In this section, we present the baseline results achieved by our tool. For the spectral data, we employed a Convolutional Neural Network (CNN) using 1-D convolutional filters, as they are well-suited for handling this type of data. Our approach involves repeated convolution and ReLu blocks, with a max-pooling operation at the end of each block. To perform the classification, we utilize a final fully connected softmax layer. Additionally, we trained a CNN model on the imagery dataset using a 2-D convolutional architecture. This approach yielded promising results, with 63% accuracy on the spectral data and 88% accuracy on the imagery data for a multi-class problem. One notable improvement in this work is the ability of the models to handle multi-class problems, even with limited data samples. This is a significant advancement from our initial device, which was only capable of binary classification. We have confidence in the efficacy of our models and their ability to perform well on diverse classification tasks.

| Confusion matrix for spectral data with CNN | | | | Confusion matrix for image data with CNN | | |
|---|---|---|---|---|---|---|
| | Healthy | CBSD | CMD | | Healthy | CBSD | CMD |
| Healthy | **63** | 4 | 33 | Healthy | **94** | 3 | 3 |
| CBSD | 4 | **70** | 26 | CBSD | 3 | **86** | 11 |
| CMD | 22 | 22 | **56** | CMD | 8 | 6 | **86** |

## 3 CONCLUSION

We have presented a cost-effective spectrometer designed as a smartphone attachment, merging the fields of machine learning and affordable technology. Our research focuses on utilizing the captured spectrum to identify crop diseases. Initial datasets and methodologies have been presented, laying the foundation for recognizing various diseases across different crops. Future experiments aim to pilot the device on a larger scale, accommodating a wider range of diseases and crops. Additionally, we aim to explore early disease detection, aiding in proactive measures before crops show symptoms. To evaluate the performance of the device, we will compare it with a high-end spectrometer (CID Bio-Science, Inc., 2010) that was utilized in our previous study (Owomugisha et al., 2020b; 2021). By accurately diagnosing diseases and assessing their severity, this device holds great potential to enhance agricultural productivity and food security in Africa, uplifting the lives of smallholder farmers.

---

[2]https://github.com/joshard/lowCostSpecData

## 4 ACKNOWLEDGEMENT

This work was done with funding from Data Science Africa Research Award: AAIRA-2022-14, ATPS, IDRC and AI4D Africa: AI4AFS/GA/AFS-0163245214. The authors would like to thank the Directors of the Uganda National Crop Resources Research Institute (NaCRRI), for granting them permission to access crop fields and screenhouses.

## 5 DATA AVAILABILITY

Original data is avaliable here

## URM STATEMENT

The authors acknowledge that one of the authors meets the URM criteria of ICLR 2023 Tiny papers Track.

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
