# OpenReview forum: "A Light Spectrometer Device for Crop Disease Monitoring"
_ICLR.cc/2023/TinyPapers — Submitted to Tiny Papers @ ICLR 2023_

### Official Review · Reviewer_6ngP · 2023-03-31

**Confidence:** 3

**Summary Of Contributions:**

This paper proposes a low-cost solution for diagnosing plant diseases. The solution consists of a low-cost smartphone-mounted light separation device and an app that makes the diagnosis based on the spectrum of the light.

**Rating:**

Great Start (GS): a submission which meets some of the reviewing criteria but has room for improvement

**Strengths And Weaknesses:**

Strengths:
This work proposes an interesting and low-cost approach to diagnosing plant diseases. The paper provides a sufficient description of how visible light is diffracted with a simple add-on mounted on a smartphone.

Weaknesses:
However, the reader is left wondering about how the spectrum information is converted into a useful diagnosis. The paper mentions some software application, but its description is completely missing.   Therefore, the reader learns almost nothing about how the system works as a whole.

**Suggested Changes:**

1) There needs to be a section that describes how the signal from the camera is processed and how it is converted into a prediction about the presence (or absence) of a disease. For example, the image in Fig. 1 could be made smaller to fit another panel describing the software (signal processing and the model (if any)). The information about the software part of the solution proposed in the paper should be sufficient allow the reader to at least gain a conceptual (if not technical) understanding of how the system works.
2) The paper should discuss the limitations of the proposed approach. For example, which diseases can and cannot be detected, how reliably, in which light conditions etc. This additional info could be included in either in the main text or added as a short appendix.
3) Fix the formatting of references in the text paying attention to the correct use of brackets.

---

> ### Author Response · Authors · 2023-05-31
> **Signal processing, light conditions, limitations on diseases diagnosis and reference formatting**
>
> We appreciate the constructive remarks provided by the reviewer. We agree that providing additional information on signal processing and analysis would be beneficial for the readers. However, due to the constraints of the current paper's length, we have opted to refer readers to a previous study (referenced) for more comprehensive details on spectral analysis and feature relevance. We also acknowledge that the topic of light conversion is relevant to our work and will be addressed in future versions of this research. The current results section presents the diseases that our tool is capable of detecting and the conclusion highlights the future prospects of this work.
>
> There was an oversight regarding the brackets in the references in the current uploaded version. We appreciate your diligence in pointing this out, we will be updated in the final version.

---

### Official Review · Reviewer_tndf · 2023-03-31

**Confidence:** 4

**Summary Of Contributions:**

The authors claim to have developed a device at a low cost which, when used together with an unspecified smartphone application, allows for the identification of plant diseases.

**Rating:**

Needs Clarification (NC): a submission which does not meet the reviewing criteria and needs clarification for its described problem or solution

**Strengths And Weaknesses:**

There is no information about or description of the "device" that was developed.

Based on what follows "Improvement in the current design lies in the following:" I am unclear as to whether any work was done to develop something or was this paper just an idea that is yet to be developed? Is the product available? Can it be used by others? If so, how?

**Suggested Changes:**

A lot more information on the actual product and/or any machine learning pipeline is required. What is this "device" that the authors mention? How does it work? How was the price tag determined or achieved? The authors say that the design follows initial prototypes but this is very vague. More information should be presented. They also mention that the smartphone camera is used as a "receiver for light that has been absorbed from the leaf sample" but it is unclear what is doing the absorbing: the camera? the leaf? the device? What is the principle at work here?

What is the scope of the product's applicability? What diseases can it diagnose? Can it be used worldwide or only in certain geographical regions?

---

> ### Author Response · Authors · 2023-05-31
> **Device description, Machine Learning Pipeline and Future work**
>
> We appreciate the reviewer's contribution in helping us improve our work. Our primary objective was to develop an affordable device that can effectively diagnose crop diseases directly in the field. We have collected data with this device on three distinct crop diseases, as detailed in our paper. This initial dataset has also been made publicly available.  Additionally, we have presented a baseline machine learning pipeline for analyzing this data. The device's design specifically focuses on utilizing low-cost materials, and it builds upon our previous artifact, which is duly referenced. In the updated conclusion, we have acknowledged the limitations of the device and outlined future directions for scaling the tool to diagnose a broader range of diseases across various crops.

---

### Official Review · Reviewer_rcyr · 2023-04-02

**Confidence:** 4

**Summary Of Contributions:**

Paper describes an low cost approach of learning representation of crops. Representation is learned in the form of wavelength. Portable device is created that is integrated into a smartphone. App is created in smartphone making it easy for farmers to use.

**Rating:**

Great Start (GS): a submission which meets some of the reviewing criteria but has room for improvement

**Strengths And Weaknesses:**

Strengths:

- Paper does include a discussion of relevant literature.
- Design of the tool/application is based on two papers. My understanding is that tool designed on the basis of these two papers should be able to make things easy for farmers and be low-cost. Keeping that in mind, I find correctness a strong point.

Weaknesses:

- I don’t see any findings mentioned in the paper.
- Design of the application is on the basis of two papers. I think readers can reproduce some work by reading those papers. However, more details regarding the design of the application will also be required to completely design the application end to end. Keeping that in mind, I find reproducibility a weak point.

**Suggested Changes:**

- In the introduction, it would be good to mention the cost of devices that are more expensive than one that has been created in this paper. It gives the reader a good idea about how low-cost this solution actually is.
- In Figure 1, the Wavelength graph is super important as that is the data point being measured by the device. However, the diagram is not that clear. It would be good to add a better diagram that clearly shows the X-axis.
- It is mentioned in the paper that the system has been developed with a mobile application and thus making it user-friendly. I think it is worth adding some details regarding features of this mobile application that is making it user-friendly for farmers.

---

> ### Author Response · Authors · 2023-05-31
> **Cost of device, Data availability and user-friendly**
>
> We thank the reviewer for the thoughtful feedback. In the updated version, we have mentioned the cost of the device ($5 - 8). Additionally, we have made the dataset captured with this tool available, allowing readers to experiment with it. One of the key advantages of our device is its user-friendliness, enabling farmers to easily collect data and identify crop diseases directly in the field.

---

### Comment · Area_Chair_tZZP · 2023-06-06
**Archival Criterion Check**

Since authors didn't provide a revised version of the paper, it doesn't meet the criterion for archival.

---

### Meta-Review · Area_Chair_tZZP · 2023-04-07

**Recommendation:** Invite to revise
**Confidence:** 5

**Metareview:**

The paper proposes an interesting low-cost solution for crop disease monitoring. However, as many reviewers mentioned, the progress of the proposed device is unclear. The strengths and weaknesses from the reviewers' comments are summarized as follows.

Strengths:
1. The paper includes a discussion of relevant literature.
2. It presents a low-cost approach to diagnosing plant diseases.
3. The design of the tool/application is promising.
4. The proposed solution has the potential to make things easier for farmers.

Weaknesses:

1. The paper lacks findings to support its claims.
2. There is insufficient information about the device itself.
3. Details about the product's progress status are unclear.
4. The explanation of how the spectrum is diffracted by the device and how the software reaches a diagnosis using the spectrum information is lacking.
6. Reproducibility of the application is a concern due to the lack of details provided.

**Summary:**

The paper introduces a low-cost, smartphone-integrated solution for diagnosing plant diseases using a light separation device and a user-friendly app, making it accessible for farmers.

**Comments And Feedback To The Authors:**

The authors proposed an interesting idea for monitoring crop disease, and the related works discussed in the paper are commendable. However, reproducibility is a major concern shared by the reviewers. The work would be much more compelling if more details about the device are discussed and comparisons with more expensive devices are conducted.

**Reason For Not Giving A Higher Recommendation:**

The major concern shared by the reviewers is that the paper lacks reproducibility due to the limited information provided about the device and application itself. Although the proposed idea has the potential to solve a real-world problem, the lack of detail regarding how the idea works serves as a strong reason for the authors to revise their work.

**Reason For Not Giving A Lower Recommendation:**

N/A

---

> ### Author Response · Authors · 2023-05-31
> **Feedback and suggestions from reviewers**
>
> We thank the reviewers for the remarks and very constructive and thorough feedback. We did our best to implement all comments and suggestions. Information about the high-end spectrometer has been captured both in introduction and conclusion. Furthermore, we have presented initial datasets obtained from the tool and outlined the methodology for analyzing spectral data to identify diseases.

---

### Decision · Program_Chairs · 2023-04-08

Revision accepted; invite to archive